# Black carbon scavenging by low-level Arctic clouds

**Paul Zieger** [1,2] ✉, **Dominic Heslin-Rees**[1,2], **Linn Karlsson**[1,2], **Makoto Koike** [3], **Robin Modini** [4] & **Radovan Krejci** [1,2]

Black carbon (BC) from anthropogenic and natural sources has a pronounced climatic effect on the polar environment. The interaction of BC with low-level Arctic clouds, important for understanding BC deposition from the atmosphere, is studied using the first long-term observational data set of equivalent black carbon (eBC) inside and outside of clouds observed at Zeppelin Observatory, Svalbard. We show that the measured cloud residual eBC concentrations have a clear seasonal cycle with a maximum in early spring, due to the Arctic haze phenomenon, followed by cleaner summer months with very low concentrations. The scavenged fraction of eBC was positively correlated with the cloud water content and showed lower scavenged fractions at low temperatures, which may be due to mixed-phase cloud processes. A trajectory analysis revealed potential sources of eBC and the need to ensure that aerosol-cloud measurements are collocated, given the differences in air mass origin of cloudy and non-cloudy periods.

Atmospheric black carbon (BC) is a primary aerosol formed from incomplete combustion, either anthropogenic or natural. BC, as a light-absorbing aerosol, can exert a significant influence on the Arctic atmospheric energy balance; thus, mitigation of BC, a short-lived climate forcer, has been reported to help offset warming[1–4]. In addition to direct radiative forcing by absorbing radiation in the atmosphere, the deposition of BC on snow and ice results in a changed surface albedo and therefore in an increase in energy absorption at the surface[5–8]. The ability of aged BC to serve as cloud condensation nuclei (CCN)[9,10] constitutes some of the other mechanisms by which BC contributes to climate radiative forcing in the Arctic. BC sources within the Arctic are rather limited and most BC observed in the Arctic undergoes long-range transport from lower latitudes[11]. Emissions from Eurasia contribute the most to surface concentrations[12–14], with typical long-range transport times of BC to the Arctic on the order of several days[15,16].

The ability of aerosol particles to act as cloud condensation nuclei is controlled by the overall governing meteorological conditions that determine the maximum supersaturation in clouds[17]. This is followed by the critical supersaturation that governs the activation of aerosol particles, which is controlled by the respective particle size, mixing state, and chemical composition[10,18–21]. BC in the Arctic still contributes less in terms of particle number and thus to CCN than other natural primary or secondary aerosol sources[22], however, the incorporation of BC into cloud particles is an important aspect since it determines the atmospheric lifetime of BC and its deposition onto snow or ice[23]. Freshly emitted BC particles exhibit hydrophobic characteristics; however, during transport to the Arctic, the aerosol ages, growing through particle coagulation and condensation of gas phase species, thus becoming more hygroscopic[24,25] and more likely to act as CCN. Recent studies have shown the importance of simulating the correct supersaturation for the Arctic BC budget[26], however, uncertainties in simulating the correct corresponding updraft velocity remain a challenge, especially for the Arctic[27].

BC can be incorporated into cloud droplets via scavenging, of which there are two different mechanisms: nucleation and impaction scavenging[20,28]. Within this work, we will use the general term BC scavenging, since it is not possible to differentiate between these two processes with the experimental techniques used here. However, it should be kept in mind that BC scavenging is dominated by nucleation scavenging[20]. Once the BC is scavenged, it can then be removed

[1]Department of Environmental Science, Stockholm University, Stockholm, Sweden. [2]Bolin Centre for Climate Research, Stockholm University, Stockholm, Sweden. [3]Department of Earth and Planetary Science, University of Tokyo, Tokyo, Japan. [4]Laboratory of Atmospheric Chemistry, Paul Scherrer Institute, Villigen, Switzerland. ✉e-mail: paul.zieger@aces.su.se

completely from the atmosphere as a result of precipitation. It should be made clear that the following text refers to the incorporation of BC into cloud hydrometeors, and does not concern itself with the removal of scavenged aerosol.

The large uncertainties surrounding the impacts of the indirect effects of aerosols on clouds (e.g. aerosol acting as CCN) mean that an estimate of the net impact of BC in the Arctic is still uncertain. Here, we present results from continuous parallel observations of total (whole-air) and scavenged BC concentrations in- and outside of low-level Arctic clouds. Our unique observations span 4 years and were conducted between 2015 and 2019 at the Zeppelin Observatory (ZEP), located on the Norwegian archipelago of Svalbard. Together with an air mass origin analysis, cloud water content (CWC) measurements, and meteorological parameters, we address the following research questions: (i) To what extent is BC scavenged by low-level Arctic clouds? (ii) How does BC scavenging change with seasons? (iii) Is there a difference in BC sources relevant to the clouds at Zeppelin Observatory? (iv) How does BC scavenging depend on other key meteorological and cloud parameters?

## Results and discussion
### The annual cycle of eBC in- and outside clouds
The annual cycle of equivalent black carbon (eBC) concentrations for cloud-free periods at Zeppelin Observatory, shown in Fig. 1a, reveals the typical seasonal cycle with a maximum in early spring (March), due to the Arctic haze phenomenon[29], followed by cleaner summer months with very low eBC concentrations. Note that arithmetic mean values in

Fig. 1a are generally above the monthly median values due to the contribution of sporadic long-range transport events of polluted air to the site. This annual behavior is governed by the seasonality of the respective sources and sinks of particles[30,31]. The observed cycle is very similar to previous long-term observations of eBC[32] concentrations for the years 1998–2007, with lower mean values (approximately a factor of two) most likely due to the generally decreasing trends of eBC concentrations at Zeppelin Observatory and the Arctic in general[14,33–35]. However, the seasonal cycle and magnitude of eBC as shown in Fig. 1a agree very well with the observed seasonal cycle reported by Sinha et al.[34] for the years 2006 to 2015. Figure 1a also shows that both MAAP instruments used in the presented study to derive the eBC concentrations of whole-air and cloud residuals, and later used within this work to derive scavenged fractions of BC, agree well when measuring at the same inlet.

During cloudy periods (Fig. 1b), the whole-air eBC concentrations (consisting of both activated and interstitial particles) follow the typical seasonality but with clearly lower overall concentrations compared to non-cloudy periods (median and interquartile range, IQR, for the entire four years: 6.2 (IQR: 2.7 - 14.0) ngm$^{-3}$ for non-cloudy periods and 2.1 (IQR: 0.7 - 4.7) ngm$^{-3}$ for cloudy periods, respectively). The reason for the lower observed concentrations of eBC (around a factor of 3 for the entire data set) is that the source regions during cloudy periods are more influenced by open water (see Figs. S1a, S2, and S3 in the supplementary information (SI) and Sect. 2.3) with fewer sources of eBC emission and/or higher chances of wet removal (due to higher relative humidity) over marine areas

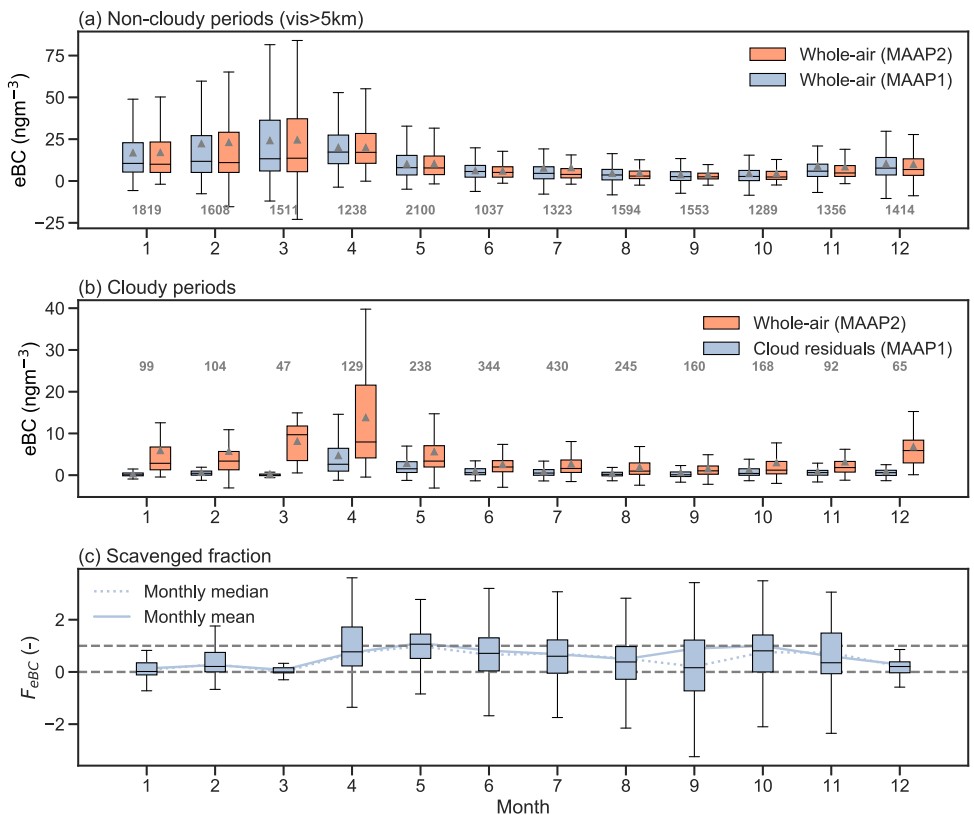

**Fig. 1 | The seasonal cycle of equivalent black carbon (eBC) in ambient and cloudy air and the resulting scavenged fraction (November 2015 until November 2019). a** Box plot of eBC concentration for ambient air (periods with visibility above 5 km) of both MAAP (multi-angle absorption photometer) instruments sampling on the whole-air inlet. **b** Box plot of eBC concentration for cloudy air (periods with visibility below 1 km and GCVI in operation). MAAP1 represents the eBC values for cloud residuals (corrected for the GCVI sampling efficiency and enrichment factor), while MAAP2 represents the eBC values for whole-air (both

cloud residuals and interstitial air). Note the different *y* axis scale in **a**, **b**. **c** Scavenged fraction $F_{eBC}$ of eBC as a box plot. The monthly mean and median values are shown as solid and dashed line, respectively. The center line of the box plot represents the median, while the triangles show the arithmetic mean of the distribution. The edges of the boxes are the quartile range and the whisker is the range of the data (defined as 1.5 times from the nearest quartile). The gray numbers give the number of hourly values contained in each box (same number of points in **c** as in **b**). Source data are provided as a Source Data file.

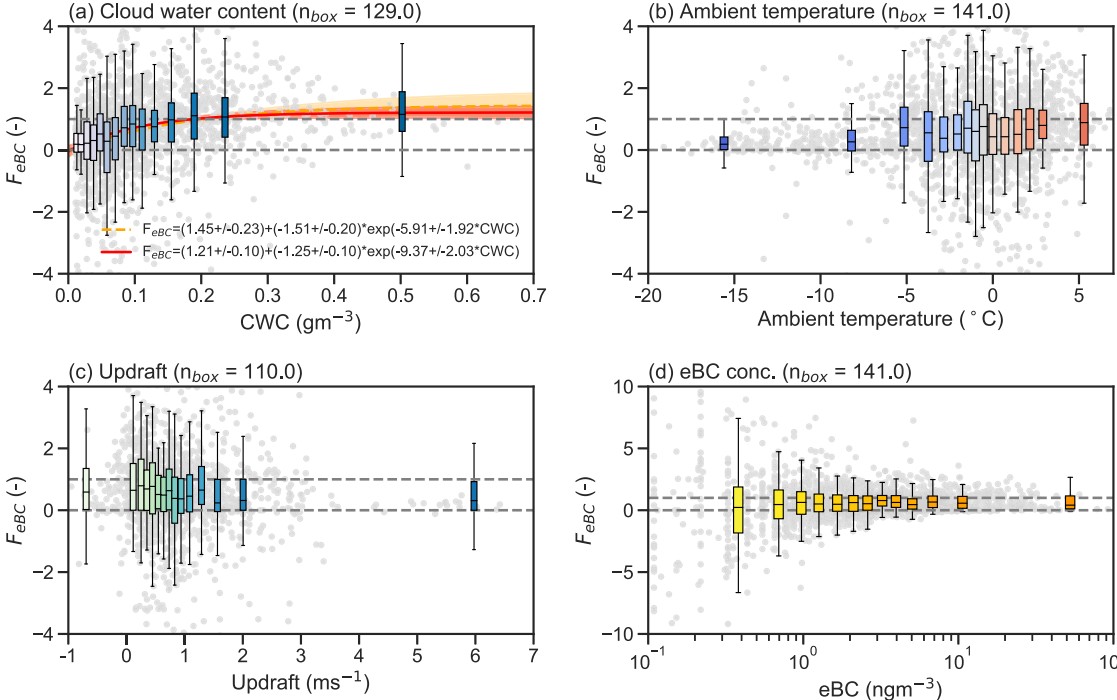

**Fig. 2 | The scaveneged fraction of equivalent black carbon binned by various other concurrent observables.** The scavenged fraction of equivalent black carbon ($F_{eBC}$) binned by **a** cloud water content (CWC), **b** ambient temperature, **c** updraft velocity and **d** whole-air equivalent black carbon (eBC) concentration. The exponential fits in **a** are shown for the 1-h mean values (orange dashed curve) and for the binned median values (red curve), respectively, together with their corresponding 95% confidence intervals (shaded area). The corresponding fit coefficients are given in the legend together with their 95% confidence intervals. The number of approximate data points (1-h values) in each box are shown above each panel. The center line of the boxes represents the median, while the extend of the boxes shows the upper and lower quartile values. The whiskers indicate the range of observed $F_{eBC}$ (defined as 1.5 times the interquartile range from the nearest quartile). The shading of the color of the box plots denotes the values of the $x$ axis. The gray dots show the underlying 1-hour mean values (**a**–**c** are limited to $F_{eBC} = \pm 4$ and **d** to $\pm 10$, respectively). The gray dashed horizontal lines are to guide the eye as the ideal range of $F_{eBC}$ is between 0 and 1. Source data are provided as a Source Data file.

along the transport to the site. The concurrently measured eBC concentration of the cloud residuals shows lower values in winter (e.g. in January with the median whole-air eBC concentration of 2.8 (IQR: 1.3–6.7) $ngm^{-3}$, compared to the median cloud-residual eBC concentration of 0.1 (IQR: -0.1–0.6) $ngm^{-3}$), indicating that not all eBC has been activated or taken up by cloud particles. Later in spring and summer, the eBC concentrations of the cloud residuals were similar to the whole-air measurements (median between 0.6 to 8.0 $ngm^{-3}$ between April and June), revealing that most of the eBC was activated into cloud droplets. The two sets of measurements (cloud residual and whole-air) present distributions that are significantly different, for all months of the year except May and October; using the Wilcoxon rank-sum test it was shown that the eBC concentrations for the whole-air inlet measurements were significantly greater than the eBC concentrations for the cloud residuals. The two sets of eBC measurements, for both May and October are not significantly different and correspond to the two peaks in $F_{eBC}$ (see Fig. 1c).

The general lower activation of eBC in winter months is further illustrated in Fig. 1c showing the scavenged fraction $F_{eBC}$ of eBC (see Eq. (2)) as box plots and as monthly mean and medians (calculated by dividing the monthly mean or median eBC concentrations of residuals by the mean or median concentrations of the whole-air). From October to March, $F_{eBC}$ gradually decreases from median values -0.8 (-80% of eBC scavenged into cloud particles) to around 0 to 0.3 meaning that only 0% to 30% of eBC is scavenged by cloud particles and the remaining 100% to 70% stays in the interstitial phase. From March onward, $F_{eBC}$ increases again towards unity (in terms of the median) which is reached in late spring (May). In summer, $F_{eBC}$ stays high with a small decrease toward the beginning of the fall. During summer, the aerosol size distribution at Zeppelin Observatory is generally dominated by Aitken-mode particles and fewer accumulation mode particles[30,36] and thus the eBC in the accumulation mode (Ohata et al.[37] found an average mass median diameter of 228 nm for BC measured at the same site) will be preferentially scavenged, also due to its increased hygroscopicity as it has undergone aging[24]. The lower $F_{eBC}$ values observed during winter could also be explained by larger fractions of accumulation mode particles in the total aerosol[38], differences in the size and mixing state of BC particles, or the importance of mixed-phase cloud processes as discussed in the next section. Figure S4 (in the SI) shows that higher values of $F_{eBC}$ are not significantly driven by increased updraft values, as removing $F_{eBC}$ values with high updraft values (above $1 ms^{-1}$ and $0.5 ms^{-1}$, respectively) keeps the monthly distribution of $F_{eBC}$ values almost unchanged (see also Fig. 2c in next section). Adachi et al.[22] observed, at the same site, the smallest ratio of residual relative to ambient carbonaceous particles during the winter season, thus observing similar seasonality.

Heintzenberg and Leck[39] determined eBC fractions in the early 1990s at the same site, comparing eBC concentrations within and outside clouds (using a $PM_1$ inlet) by means of an optical filter-based light absorption technique. As such, the study determined the ratio between interstitial (during cloudy periods) and $PM_1$ during out-of-cloud periods, which can be converted to $F_{eBC}$ (similar as done by Cozic et al.[40]) by assuming that the residual eBC concentration equals the eBC concentration of $PM_1$ minus the interstitial value. They did not use a cloud sensor to determine the in-cloud periods but used a data reduction scheme to infer the presence of clouds at the station. Heintzenberg and Leck[39] found for summer (mid-May to mid-October 1990-1992) and winter (mid-October to mid-May 1990–1992) average $F_{eBC}$-values of 0.81 and 0.77, respectively. If we calculate the median for the same monthly periods as Heintzenberg and Leck[39], we receive

scavenged fractions of 0.42 for their winter and 0.53 for their summer period, respectively. This is substantially different and probably can be explained by the different setup, the different cloud-detection scheme, and that the fractions determined by Heintzenberg and Leck[39] did not use temporarily collocated measurements and therefore did not consider differences in air mass origin during cloudy periods compared to non-cloudy periods, where the overall concentrations of eBC are different (cf. Fig. 1).

### Scavenged fraction of eBC versus cloud and meteorological parameters

Figure 2 shows $F_{eBC}$ versus CWC, ambient temperature, updraft velocity, and whole-air eBC concentration. $F_{eBC}$ shows a clear dependency on CWC, as shown in Fig. 2a. Clouds with higher CWC tend to scavenge more eBC, which was similarly observed at the Jungfraujoch high-alpine site in the Swiss Alps[40]. Two fits for both 1-h values and the binned median values of $F_{eBC}$ are added to Fig. 2a, showing an exponential increase of $F_{eBC}$ with increasing CWC (see legend for fit coefficients and confidence intervals). In Cozic et al.[40], $F_{eBC}$ values were slightly lower, reaching median of around 0.7 at 0.6 gm$^{-3}$ CWC, while $F_{eBC}$-values of around 1 (median) at Zeppelin Observatory are already reached at around 0.15 gm$^{-3}$ CWC. The larger $F_{eBC}$ values in the present study could be due to longer transport times to the Arctic and therefore a more aged and more hygroscopic aerosol that is scavenged more easily into the clouds, compared to Jungfraujoch[41,42].

The observed $F_{eBC}$-values also show two distinct regimes or populations with respect to the ambient temperature, with clearly lower scavenged fractions below around −5 °C. This observed decrease with decreasing temperature can be explained by the presence of ice and the Wegener-Bergeron-Findeisen (WBF) process. At colder temperatures, when mixed-phase clouds may be present, liquid droplets evaporate at the expense of growing ice crystals, leaving more BC in the interstitial phase (leading to a lower $F_{eBC}$). A similar decrease in $F_{eBC}$ at temperatures below around −5 °C has been also observed by Cozic et al.[40] who also attributed this effect to the WBF process.

Figure S5 (in the SI) shows the same dependence of $F_{eBC}$ versus CWC as in Fig. 2a, but for the two different temperature regimes above and below -5 °C separately. Removing the observations below −5 °C does not change much the general dependence of $F_{eBC}$ with CWC as shown in Fig. 2a, since liquid clouds dominate the data. For temperatures below −5°, however, the CWC is much smaller due to the dominance of cloud ice, and the median $F_{eBC}$ values are at around 0.2 until -0.05 g m$^{-3}$ CWC. This dependence is in line with the findings of Cozic et al.[40] (although their observed CWC extended to higher values of around 0.45 g m$^{-3}$ CWC compared to here). The relationship of $F_{eBC}$ versus CWC (Fig. 2a) is also reflected in terms of eBC concentration (see Fig. S6a in the SI) meaning that more eBC is incorporated into the cloud particles with increasing CWC. However, this relationship is not seen for the whole-air eBC concentration versus CWC (see Fig. S6b), with even slightly higher eBC concentrations at low CWC. This again indicates that more eBC stays in the interstitial phase especially in the winter, when CWC is generally lower compared to the summer (see Fig. S6c and Fig. 1b, c).

The relationships of $F_{eBC}$ versus CWC and $F_{eBC}$ versus temperature are interlinked with the seasonality of clouds observed at Zeppelin Observatory. In the winter, the observed clouds were in general thinner with lower CWC (or higher visibility) compared to the summer months (see Fig. S6c, d in the SI). Therefore, the seasonality of lower $F_{eBC}$ in the winter months (see Fig. 1c) can be explained by the fact that CWC is generally lower in the winter (Fig. 2a and Fig. S6c in the SI) and in parts due to mixed-phase cloud processes (Fig. 2b). The separation between both effects is only possible with collocated detailed cloud microphysical measurements which include the separation between cloud droplets and ice crystals.

Similarly to the previously discussed Fig. S4 (in the SI), Fig. 2c reveals that the scavenged fraction $F_{eBC}$ does not show a clear

dependency on the updraft velocity, suggesting that the activation of eBC at Zeppelin Observatory is not much influenced by local orographic effects. This also indicates that the presented results are representative of low-level clouds in the Svalbard region, making this long-term data set suitable for model improvements or model validation exercises. The higher variability of $F_{eBC}$ at very low concentrations of eBC (see Fig. 2d) also explains why $F_{eBC}$ can sometimes be above unity (see Figs. 1c and Fig. 2a–c), since extremely small and thus uncertain concentrations are used to calculate $F_{eBC}$.

### Sources of eBC in- and outside clouds

As described earlier, air masses arriving at Zeppelin Observatory where low-level clouds were observed, were associated with more of an oceanic origin as compared with cloud-free air masses (c.f. Fig. S1 and S7 in the SI). Therefore, the source region of eBC in- and outside cloud periods is expected to be different. This is further demonstrated in Fig. 3, showing that cloudy periods are characterized by very low eBC concentrations with less continental influence. Slightly elevated concentrations can be attributed to areas of the Norwegian coast most likely from an anthropogenic source[43,44]. Splitting the source maps into seasons (see Fig. S8 in the SI) gives further indication that the sources of eBC observed during cloudy periods are to a large extent linked to anthropogenic activities in the Norwegian and Northern Sea (such as oil and gas production), although it is harder to confidently assign specific source regions due to reduced data coverage for the individual seasons. During non-cloudy periods (see Fig. 3b), the source regions are different, with elevated eBC values associated with air masses originating from Eurasia and especially from central Russia; regions where fossil fuel combustion and biomass burning especially in summer, contribute to eBC in Zeppelin Observatory[45]. Figure S2 (in the SI) shows that for non-cloudy eBC, measured via the whole-air inlet, there is a clear dependence on the time that coinciding air masses spend over continental source regions, whereas for cloudy periods eBC concentrations of cloud residuals are much lower and there is no relation to the time air masses spend over continental regions; observations of cloud residuals coinciding with air masses influenced more by marine surface types have a similar eBC concentration as to air masses influenced more by the continent (see Fig. S2a in the SI), a finding also displayed in the homogeneous eBC concentrations in Fig. 3a. The clear differences in air mass origin during cloudy compared to non-cloudy periods implies that the air mass origin needs to be taken into account when using aerosol in situ observations to study cloud processes or when performing model measurement evaluation exercises using aerosol in situ data.

## Methods

Aerosol and cloud particles were sampled from November 2015 to November 2019 at the Zeppelin Observatory on Svalbard in the high Arctic. The observatory (78°54′ N, 11°53′ E) is situated on a mountain ridge at 475 m altitude, ~2 km south of the research village Ny-Ålesund, where it is largely unaffected by local pollution. For more than 30 years, the observatory has been part of various international air monitoring programs (see Platt et al.[46] for a recent review).

### Determining aerosol absorption properties of whole-air and cloud residuals

Two inlet systems were used: a whole-air inlet to sample all air (aerosol and cloud particles) and a ground-based counterflow virtual impactor inlet (GCVI) to sample only cloud particles (see Fig. S9 in the SI for a schematic setup). The whole-air inlet follows the guidelines of the World Meteorological Organization (WMO) Global Atmosphere Watch (GAW) program[47,48] that can sample cloud particles up to 40 μm in diameter at wind speeds up to 20 ms$^{-1}$. It is slightly heated to temperatures between 5–10 °C to prevent freezing. The GCVI inlet (Brechtel Manufacturing Inc, USA, Model 1205) separates large particles, here cloud droplets or ice crystals, from interstitial or non-

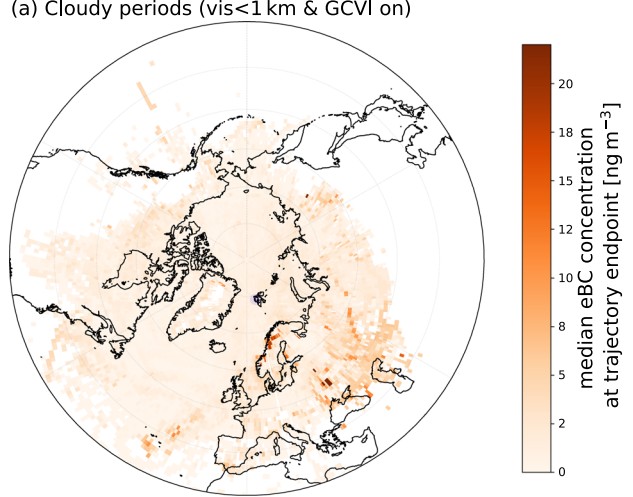

(a) Cloudy periods (vis<1 km & GCVI on)     (b) Non-cloudy periods (vis>5 km)

**Fig. 3 | Source maps of equivalent black carbon (eBC) concentration for cloudy and non-cloudy periods for the entire 4-year data set. a** Shows the eBC concentration of cloud residuals for cloudy periods with GCVI (ground-based counterflow virtual impactor inlet) sampling (visibility < 1 km), whereas **b** shows the eBC

concentration for non-cloudy periods (visibility > 5 km). During the non-cloudy periods, air masses resided for relatively longer periods of time over continental BC source regions in Europe and Russia compared to cloudy periods. Source data are provided as a Source Data file.

activated aerosol and samples uniquely the cloud particles. A detailed technical description of the CVI inlet used here can be found in Shingler et al.[49] and only a short summary will be given here. A visibility sensor is used to detect the presence of a cloud, which here is defined as visibility below 1 km for at least 5 min. Using a wind tunnel, the sampled cloud is accelerated to -120 m s⁻¹ onto the CVI tip, where a counterflow of dried air allows only large particles above 6–7 μm (aerodynamic diameter) to pass the virtual stagnation plate. The sampled particles are then dried and the remaining residuals, termed cloud residuals, are analyzed by aerosol light absorption instrumentation. The GCVI setup used here has been evaluated in detail in Karlsson et al.[38] using the measured particle size distributions and concentrations in parallel of cloud residuals and whole-air, and additionally using the collocated cloud particle measurements. The sampling efficiency of the GCVI was determined to be -0.46, which was confirmed by comparing the number concentration of accumulation mode of cloud residuals and whole-air (see Karlsson et al.[38] for details).

Two identical multi-angle absorption photometers (MAAP, Thermo Fisher Scientific, Germany, Model 5012) were used to determine the equivalent black carbon (eBC) concentration. The MAAP determines the particle light absorption coefficient $\sigma_{ap}$ (in m⁻¹) at a wavelength of $\lambda = 637$ nm[50] of particles that are deposited on a filter tape (see Petzold and Schönlinner[51] for further details). The instrument assumes a BC mass absorption cross-section (MAC$_{BC}$) of 6.6 m² g⁻¹ to estimate the eBC concentration, however, we applied a value of MAC$_{ZEP}$ = 10.6 m²g⁻¹ based on the recent and site-specific findings of Ohata et al.[52]:

$$\text{eBC} = \frac{\sigma_{ap}(\lambda)}{\text{MAC}_{ZEP}(\lambda)}. \qquad (1)$$

All $\sigma_{ap}$ were corrected by a factor of 1.05 to account for differences in wavelength between the wavelength stated by the manufacturer (670 nm) and the operational wavelength (637 nm) as recommended by Müller et al.[50] All eBC values are given at ambient pressure and temperature. During periods without GCVI operation (e.g., cloud-free periods), both MAAP instruments were sampling at the same whole-air inlet using an automated three-way valve that connected one MAAP (MAAP1) to the GCVI line when the latter was in operation (see Fig. S9 in the SI). This allowed us to constantly intercompare both instruments and identify possible malfunctioning. The MAAP that determined the

eBC concentration of the whole-air (MAAP2) was sampling at a flow rate of 16.6 lpm, while MAAP1 operated at a lower flow of 5 lpm. Both instruments were set to a time resolution of 1 min.

Optical filter-based measurements are sensitive to abrupt changes in relative humidity and temperature. As a conservative approach and based on the comparison of the mean and median difference in eBC concentration (see Fig. S10 in the SI), the first 15 min of MAAP1 data, after the GCVI was turned on or off, were disregarded. Based on the work by Asmi et al.[53], hourly arithmetic mean values were calculated (requiring a minimum of at least 30 1-min values per hour). This resulted in 4.2% and of the MAAP1 and 2.2% of the MAAP2 data, respectively, being below the detection limit of 0.012 Mm⁻¹ as determined by Asmi et al.[53] for the same type of instrument (operated at the Arctic site of Pallas, Finland). The values below the limit of detection were included in the monthly or binned data to avoid a positive bias. In total, 2158 hourly averages of in-cloud data (MAAP1 and MAAP2) with ambient visibility below 1 km were collected throughout the four years with generally more data availability during the summer months (see Fig. S11 in the SI). Data coverage for cloud-free periods, here defined as visibility above 5 km and GCVI turned off, was generally more and -17,842 hourly averages of reliable data were collected by both MAAP1 and MAAP2 (see Fig. S11).

The scavenged fraction of eBC, $F_{eBC}$, was then calculated using the hourly mean eBC values of the cloud residuals divided by the corresponding values of the whole air:

$$F_{eBC} = \frac{\text{eBC}_{cloudres.}}{\text{eBC}_{whole-air}}. \qquad (2)$$

Note that the eBC concentrations of the cloud residuals measured behind the GCVI were corrected for the enrichment factor and the GCVI sampling efficiency (see above).

**Auxiliary measurements**
An ultra-sonic anemometer (Metek GmbH, Germany, Model Sonic-3 Omni) was used to determine the three-dimensional wind components at 1 Hz time resolution. It was placed slightly elevated about 1 m above and near the GCVI inlet (see photo in Fig. S9 in the SI).

Cloud water content (CWC) was calculated from the cloud particle size distribution for radii between 1.5 and 23.5 μm measured by a fogmonitor (Droplet Measurement Technologies Inc., USA, model FM-

120). The instrument is located on the terrace of Zeppelin Observatory around 4 m lower than the GCVI. More technical details on the FM-120 can be found in Koike et al.[54] Since no information is available on the shape of cloud particles, we assumed that the particles are spherical water droplets to calculate CWC. Data were averaged for the sampling time of the MAAP measurements.

## Source analysis

An air mass back trajectory analysis was performed to determine the origin of eBC in- and outside of clouds. An ensemble of 27 10-day back trajectories, arriving at the same latitude and longitude of Zeppelin Observatory with an initialized height of 250 m to be within the mixed layer, were calculated with an hourly resolution, using the Hybrid Single-Particle Lagrangian Integrated Trajectory model HYSPLIT (V5.2.1)[55,56] for the entire four years of observations. The ensemble of back trajectories is setup by defining a 3D dimensional grid box around the observatory, in which the meteorological data associated with the starting point is shifted, but not the initial position. The default grid offset of 1 grid point in the horizontal and 0.01 sigma units (-250 m) in the vertical is used. As meteorological input data, the Global Data Assimilation System (GDAS, https://www.ready.noaa.gov/archives.php, last accessed 2021-11-21) was used. The calculated back trajectories provide, in addition to position and altitude, also the height of the mixed layer along each back trajectory. A 1° × 1° grid centered around the observatory (acting as the new north pole) was defined, and each back trajectory endpoint was mapped onto the grid. Each cell along the back trajectory (endpoint) was assigned the 1-hour mean eBC value measured at the observatory on arrival of the air mass, if the air resided within the mixed layer. In addition, the residence time within the mixed layer was calculated per grid cell. The median eBC concentration per grid cell was computed (requiring at least 10 data points per grid cell for the median eBC concentration), and in the case of calculating residence time the number of hours was summed. These calculations were done separately for the cloudy and non-cloudy periods, as well as for individual seasons. The figures for the surface residence time were later normalized by the maximum time within the summed grid. All geospatial figures were made using the Python package Cartopy (v0.18.0)[57]

## Data availability

The processed eBC, GCVI, and updraft data are available at Karlsson et al.[58] The processed air temperature data is available at the EBAS database (https://ebas-data.nilu.no). The processed cloud in situ data are available at Koike, M., M. Shiobara, S. Ohata, N. Moteki, T. Mori, 2020, Cloud particle concentration data obtained by in situ measurements at Mt. Zeppelin, 2.00, Arctic Data archive System (ADS), Japan, https://ads.nipr.ac.jp/dataset/A20200309-001 Source data are provided with this paper.

## Code availability

All code behind the main figures is available on Zenodo (https://zenodo.org/badge/latestdoi/678923051).

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

## Acknowledgements

The authors thank research engineers Tabea Henning, Ondrej Tesar, Kai Rosman, and Birgitta Noone from the Department of Environmental Science at Stockholm University (ACES) and the staff from the Norwegian Polar Institute (NPI) for their on-site support. NPI is recognized for its substantial long-term support in maintaining measurements at the Zeppelin Observatory. We thank Ove Hermansen and Wenche Aas (Norwegian Institute for Air Research, Norway) for providing ambient temperature data from Zeppelin Observatory. We thank Roxana Cremer (ACES) for her help with the back trajectory calculations. The authors also thank Martin Gysel and Barbara Bertozzi (PSI) for their valuable discussions. This research was supported by the Knut och Alice Wallenbergs Stiftelse (grant no. 2016.0024, R.K.), the Naturvårdsverket (Miljöövervakning, R.K.), the Svenska Forskningsrådet Formas (grant no. 2016-01427, R.K.), and the Swedish Research Council (grant no. 2018-05045, P.Z.), and the the Research Council of Norway (access grants as part of Project 291644, Svalbard Integrated Arctic Earth Observing System-Knowledge Centre, operational phase, R.M.). This project has received funding from the European Union's Horizon 2020 research and innovation program under grant agreement no. 101003826 (P.Z.) via project CRiceS (Climate Relevant interactions and feedbacks: the key role of sea ice and Snow in the polar and global climate system). This research was also supported by the MEXT of Japan for the Arctic Challenge for Sustainability II (ArCS II, JPMXD1420318865, M.K.) project, JSPS KAKENHI Grant Number JP22H01294 (M.K.), and the Environment Research and Technology Development Fund (JPMEERF20202003 and JPMEERF20232001, M.K.) of Environmental Restoration and Conservation Agency.

## Author contributions

P.Z. and R.K. conceived the study and performed measurements. M.K. performed cloud in situ measurements. D.H.-R. performed trajectory

calculations. P.Z., D.H.-R., L.K., and M.K. analyzed data. R.M. provided detailed input to the discussion. P.Z. and D.H.-R. wrote the manuscript with input from all co-authors.

## Funding

## Competing interests
The authors declare no competing interests.
