## [Peer Review File · Nature Communications]

Black carbon scavenging by low-level Arctic cloudsReviewer #1 (Remarks to the Author):

I commend the authors on taking on such challenging measurements to determine the fraction of BC mass that is contained within near-surface cloud droplets in the high Arctic. They interpret their results to indicate a clear seasonality to this fraction, from low values in winter to higher values in summer. They then tie these observations to some concurrent properties of clouds (cloud water content likelihood of freezing) along with the influence of source region on the likelihood of clouds. While I appreciate the effort made, I have numerous concerns regarding the measurements, or specifically the interpretation of the measurements and the complete lack of any consideration of uncertainties. This could be an interesting paper if the results prove robust, but I need to be convinced that the results are indeed robust; as currently written, I am not. I do not support the publication of the current version of the manuscript. Whether a revised version will sufficiently address my concerns, and I can only assume the concerns of other reviewers, remains to be seen.

My various comments follow below. As line numbers were not provided, I hope that the authors are able to discern to which aspects my comments apply.

If I understand correctly, the transmission method used here cannot result in negative eBC values. Thus, the values should not be normally distributed. Instead, they are most likely log-normally distributed, in which case geometric averages would be better to use than arithmetic averages. The authors might consider making this change.

This work would certainly benefit from some sort of reasonably statistical analysis. The authors are working at the limits of the instrumental detection limits. Consider that on Page 4 they note that "The higher variability of FeBC at very low concentrations of eBC (see Fig. 2d) also explains why FeBC can sometimes be above unity (see Fig. 1c and panel a-c in Fig. 2), since extremely small and thus uncertain concentrations are used to calculate FeBC." Yes, exactly. Fig. 2d indicates that individual FeBC values both <0 and >1 are found even at the highest eBC concentration. This doesn't surprise me, given the low concentrations. And just because values <0 or >1 are found doesn't mean that the results are inherently problematic. But it does mean that the authors need to actually consider the uncertainties in some formal manner. Here, the precision, and the relative precision, should be much more important than the accuracy.

Fig. 1 is a key figure in the manuscript, underpinning the entire story. I therefore am surprised to see that the authors have, it seems, discarded the vast majority of the scavenged fraction results determined from the median values. This is because there are really only two months (March/April) during which the whole-air eBC $>$ cloud residual eBC. This contrasts with the mean values, for which the whole-air eBC is generally \geq the cloud residual eBC. This raises a key question: which is more appropriate to use? The mean or the median? There are reasons to think that either is appropriate, although for a multi-year dataset such as this I would typically consider median values more robust, as mean values can be very sensitive to extreme events. So, to my previous question, how do the results change if the geometric mean values are used instead? Also, are the conclusions robust to the exclusion of extreme values? Are the distributions normal? (Most likely not, given the substantial mean-median differences.) Given the behavior of the medians, I need to be convinced that the conclusions based on the mean values are robust. Or put differently, the authors need to relate what they observe in the mean to how they interpret things in the median throughout much of the manuscript (i.e., based on typical seasonal behavior, not extremes. As it is, the authors' are focusing on the behavior of extreme events out of the year rather than on the typical behavior. They note as such on Page 2 but still this is an important consideration in terms of what any of this means for Arctic climate. It also has implications for how the authors consider data in Fig. 2, where they seem to include all data points, including those that they just said are basically unreliable for the median. I find this overall a major weakness.

The abstract notes that "An understudied aspect is the role of BC in the formation of low-level clouds in the Arctic." To nit-pick, this study does not address this question. It addresses whether BC is found within cloud droplet residuals when considered on a mass basis. This does not say anything about what role the BC, as opposed to other soluble components, play in controlling the formation of low-level Arctic clouds. A more accurate statement that would tie to what the current study shows is "An understudied aspect is the extent to which BC is present in low-level clouds in

the Arctic." Similarly, the comes out in the first paragraph, where the authors mention "the ability of aged BC to serve as cloud condensation nuclei." Perhaps this is semantics and just a pet peeve on my part, but the question is ultimately whether the BC "serves" as a CCN or whether it is just along for the ride with the other, more soluble material the contributes to the aging process. In other words, take away the BC and would you still have the CCN? Similarly, the met conditions ultimately govern the ability of any particle to act as a CCN, not just BC (first sentence, second paragraph). In part, I'm getting at the point of whether it is important that BC serve as CCN from a perspective of the clouds, or if it is important more for the lifetime of the BC and its deposition onto snow and ice. The authors might consider making this distinction in a more nuanced manner.

The conclusions regarding the scavenged fraction in Fig. 1c rely entirely on the mean values of eBC. Yet, in Fig. 2 only median values of the scavenged fraction are shown. Why this disconnect? For consistency shouldn't one use mean values? Also, I do not think it is fair to say that the "whiskers show the range of observed eBC" when it is evident that the scales are set to cut off the full range of values.

The authors show there is a relationship between the scavenged fraction and eBC, with some hint that this is an inverse relationship. Is there a relationship between eBC and cloud water content?

The authors note that there are 2,207 hours of in-cloud data with visibility <1 km collected. In Fig. 2 it is indicated that there are N = 709 number of data points for scavenging fraction data. What is the relationship, then, between these 2,207 and 709 values? What constitutes a "data point?" And how does this relate to the statement in the methods that "The scavenged fraction of eBC, FeBC, was then calculated using the monthly mean eBC values of the cloud residuals divided by the corresponding values of the whole-air." There are only four years of data considered, so this would presumably mean a maximum of 48 "data points" where a data point is taken as a monthly mean.

The fewer number of "Data points" for the CWC and updraft periods in Fig. 2 versus the ambient temperature and eBC data points indicates that there was incomplete sampling. This is fine, and not surprising for a multi-year record. However, the authors might show as a supplemental the actual instrumental coverage to illustrate which of the 709 data points for Fig. 2b and Fig. 2c include the 590 CWC data points, and where there is not overlap. What sort of biases might this impart?

The authors state on Page 1 that "The removal of BC from the atmosphere can occur through dry or wet deposition. Wet deposition describes the process of aerosol activation and subsequent removal by precipitation, also termed nucleation scavenging." Here, I emphasize the phrase "subsequent removal by precipitation." This work does not consider this key step. It only considers the fraction of BC that is detected in cloud residuals. Given the authors' own definition of scavenging, I strongly encourage a revision to the title, abstract, terminology of FeBC (as the scavenging fraction), and really the language throughout to make clearer what they have actually characterized, which is the fraction of BC that is present in low-level clouds. This is not the same as the scavenging fraction. Should they want to refer to this as the "potential scavenging fraction" or some other such terminology I would be okay with this. But what they do not characterize is the scavenging. Scavenging must also consider the actual removal from the atmosphere term.

Page 1: How do "large uncertainties" translate to an estimate "missing?" To me this does not logically follow. The large uncertainties mean that estimates of the impact are uncertain. Not that such estimates are "missing."

Page 2: The authors note that "However, the seasonal cycle and magnitude of eBC as shown in Fig. 1a agrees very well (also with respect to magnitude) with the observed seasonal cycle using more recent and advanced observations by Sinha et al." I do not follow this, in particular the "more recent" statement. First, the Sinha paper is from 2017 yet the work here encompasses measurements from 2015-2019. So how is Sinha et al. "more recent?" Also, how are the Sinha et al. measurements "more advanced?" They too use filter-based absorption measurements. A MAAP, as used here, is arguably more "advanced" than a PSAP or COSMOS considered by Sinha et al.

Page 2: The authors note that "During cloudy periods (Fig. 1b), the whole-air eBC, consisting of

activated and interstitial (non-activated) particles, follows the same typical seasonality..." and then report mean values. But in the median the values seem to be basically, or very near zero for all cloudy periods with the exception of two months. Why focus on the mean and not the median here? As to a previous comment, wouldn't the means emphasize extreme events?

The comparison to Heintzenberg and Leck (1994) would be strengthened if the authors here used their higher time-resolution measurements to calculate the equivalent in-cloud/out-of-cloud fraction as determined by H&L. Are different results obtained? This should be as straightforward as calculating the ratio between the whole-air values from MAAP 2 in Fig. 1b and Fig. 1a. Is there a seasonal variation that can explain the H&L results?

How would the authors define a "clear dependence" (or lack thereof) as they use it with reference to the updraft velocity-scavenged fraction relationship (Page 4) or the FeBC-CWC relationship, or any other relationship? Is there some statistical metric?

The authors compare their results to those from Jungfraujoch, another remote site. However, the mean eBC concentrations in Jungfraujoch are about an order of magnitude higher than here. So, while such measurements are still challenging, the importance of a careful uncertainty assessment is different. The differences in the FeBC values between the studies could be, ultimately, negligible once uncertainties are accounted for. I am skeptical that any of the relationships shown in the current study are "real" and not simply a function of the parameter of interest (CWC or air temperature) (anti)correlating with eBC. I need to be convinced that the relationships observed are real.

On page 5 the authors state at the end of a sentence that "for cloudy periods, eBC is removed regardless of the surface type." It is not clear how this follows from the previous part of the sentence. Where does the "removal" aspect come from?

Reviewer #2 (Remarks to the Author):

Review of "Black carbon scavenging by low-level Arctic clouds" by Ziegler et al. (read abstract and introduction one more time)

Below I address the questions raised by Nature Communications about the submitted manuscript. "Q" represents the question, and "A" represents the answer.

Q1: What are the noteworthy results?

A1: The authors present the first direct long-term observational study of the scavenging of black carbon (BC) by clouds. The authors collected an impressive four years of data allowing for seasonal cycles and dependence on several factors to be elucidated. From these studies, the authors conclude that the scavenging efficiency of BC by clouds was positively correlated with the cloud water content and depended on the ambient temperature.

Q2: Will the work be significant to the field and related fields?

A2: Yes. This unique dataset will be useful for model improvements or model validation exercises. In other words, these measurements should lead to better models and predictions of the effect of BC on the Arctic climate.

Q3: How does it compare to the established literature? If the work is not original, please provide relevant references.

A3: As indicated by the authors, a previous study by Cozic et al. 2007, ACP, showed that the scavenging efficiency of BC was positively correlated with the cloud water content and depended on ambient temperature. The previous study by Cozic et al. was carried out at a European mountain site. The current study shows a similar behavior but for the Arctic region. The authors stated in the Conclusion that "the scavenging fraction of BC increased as expected with the

increasing cloud water content." In other words, the observations are expected but critical to confirm for the Arctic, which is experiencing rapid climate change.

Q4: Does the work support the conclusions and claims, or is additional evidence needed? - Are there any flaws in the data analysis, interpretation, and conclusions? - Do these prohibit publication or require revision?- Is the methodology sound? Does the work meet the expected standards in your field?

A4: The methodology is sound, and the work is of very high standards. The manuscript is also very well written, and the work supports the conclusions and claims.

Below are additional comments for the authors to consider.

1) Page 2, last paragraph. "The fractions determined by Heintzenberg and Leck did not use temporarily collocated measurements and therefore did not consider differences in air origin during cloudy periods compared to non-cloud period, where the overall concentrations of eBC are different." I had to go and look up the paper by Heintzenberg and Leck to understand the difference between these measurements and the new results by Zieger et al. The authors may want to rewrite this paragraph to clarify it for the non-expert.

2) Beginning of Section 4. "Therefore, the source region of eBC in and outside cloud periods is expected to be different. This is further demonstrated in Fig. 3 showing that cloudy periods are characterized by very low eBC concentrations of mainly marine origin". What features in Figure 3 lead to this conclusion? From Fig. 3a, the median eBC concentration at trajectory endpoints is highest over land. I agree there is more red color over land in 3b vs. 3a, but land vs. marine in 3a is not clear to me.

Reply to reviewers for manuscript “Scavenging of black carbon in low-level Arctic clouds”

Paul Zieger^{1,2}, Dominic Heslin-Rees^{1,2}, Linn Karlsson^{1,2}, Makoto Koike³, Robin Modini⁴, and Radovan Krejci^{1,2}

¹Department of Environmental Science, Stockholm University, Stockholm, Sweden

²Bolin Centre for Climate Research, Stockholm University, Stockholm, Sweden

³Department of Earth and Planetary Science, University of Tokyo, Tokyo, Japan

⁴Laboratory of Atmospheric Chemistry, Paul Scherrer Institute, Villigen, Switzerland

Correspondence: paul.zieger@aces.su.se

We thank both reviewers for their constructive and helpful comments, which helped to improve the clarity of our manuscript. Our replies are given below in blue (changes in the text are given in italics). We apologise for the late reply which was caused by the absence of the main author (PZ), who was occupied with extensive field work during April, May and June 2023.

1 Comments and reply to Reviewer #1

5 I commend the authors on taking on such challenging measurements to determine the fraction of BC mass that is contained within near-surface cloud droplets in the high Arctic. They interpret their results to indicate a clear seasonality to this fraction, from low values in winter to higher values in summer. They then tie these observations to some concurrent properties of clouds (cloud water content likelihood of freezing) along with the influence of source region on the likelihood of clouds. While I appreciate the effort made, I have numerous concerns regarding the measurements, or specifically the interpretation of the
10 measurements and the complete lack of any consideration of uncertainties. This could be an interesting paper if the results prove robust, but I need to be convinced that the results are indeed robust; as currently written, I am not. I do not support the publication of the current version of the manuscript. Whether a revised version will sufficiently address my concerns, and I can only assume the concerns of other reviewers, remains to be seen.

15 My various comments follow below. As line numbers were not provided, I hope that the authors are able to discern to which aspects my comments apply.

Thanks for the critical but constructive comments. We apologise for the previously missing line numbers, which are now added to the revised manuscript. Following the comments of the reviewer, we have completely revised our data processing, tested and implemented the various suggested approaches, performed additional statistical tests, and have substantially improved the presentation and discussion of our results. In particular, we repeated the data processing following the work of
20 Asmi et al. (2021) who determined the level of detection for various aerosol absorption instruments at the Arctic Pallas station (incl. the instrument used within this work) and changed the presentation of the results in the main figures. As described in

more detail below, the overall message has not changed and we hope that we can now convince the reviewer and reader about the robustness of our main observations and their interpretation.

25 If I understand correctly, the transmission method used here cannot result in negative eBC values. Thus, the values should not be normally distributed. Instead, they are most likely log-normally distributed, in which case geometric averages would be better to use than arithmetic averages. The authors might consider making this change.

30 The transmission method used here can indeed result in negative values, especially at very low concentrations. The MAAP, however, is much less influenced by negative values than other filter-based absorption instruments like the commonly used aethalometer due to its collocated light scattering measurements (Petzold and Schönlinner, 2004). These negative values have to be included to not introduce a positive bias when averaging. In the revised version, we now use hourly arithmetic mean values of eBC following the work of Asmi et al. (2021) and use box plots for monthly averages and the binned data (see main figures 1 and 2). With this approach, only 2.2% of the MAAP2 data and 4.2% of the MAAP1 data were below the limit of detection (LOD). We have not removed the values below the LOD when calculating monthly averages or when binning the data.

35 To describe our improved data processing, we added to the method section: *“Based on the work by Asmi et al. (2021), hourly arithmetic mean values were calculated (requiring a minimum of at least 30 1-min values per hour). This resulted in 4.2 % of the MAAP1 and 2.2 % of the MAAP2 data, respectively, being below the detection limit of 0.012 Mm^{-1} as determined by Asmi et al. (2021) for the same type of instrument (operated at the Arctic site of Pallas, Finland). The values below the limit of detection were included in the monthly or binned averages to avoid a positive bias.”*

40 The new approach in using 1-hourly data resulted in slightly lower data coverage of 2158 hourly averages of in-cloud data (before 2207 hours) and 17 842 out-of-cloud data (before 18 311 hours), which we added to the revised manuscript.

As recommended by the reviewer, we have tested the geometric mean and the results are shown in Fig. 1 below. For the geometric mean, we can only use values above zero. As expected, one can see that the arithmetic mean (here just called mean) is higher compared to the monthly median and geometric mean, due to the influence of long-range transported pollution events. Interestingly, the monthly median and geometric mean compare very well to each other for the non-cloudy cases. For the scavenged fraction, the geometric mean is higher than the median, which is not surprising as only values above 0 can be used (which is unreasonable because of eBC in the Arctic can be around 0). In the revised version of the manuscript, the geometric mean is not really needed, since we now show box plots of the entire data (with a better and more pronounced focus on the median and the spread of the data) but also show the arithmetic mean values as small triangles for the eBC concentrations (and point to the fact of outliers due to long-range transport, which is an important fact to keep in mind). In addition, in the revised main Figure 1 (see Fig. 2 below), we now show the scavenged fraction as (i) boxplots and (ii) calculated from the monthly mean and median of eBC concentration. Despite the large variability of the 1-hourly calculated F_{eBC} -values (which is reasonable due to the very low eBC concentrations), both calculated average values agree very well and show the annual cycle of decreased scavenged fraction in the winter and increased values in the summer.

55 This work would certainly benefit from some sort of reasonably statistical analysis. The authors are working at the limits of the instrumental detection limits. Consider that on Page 4 they note that “The higher variability of FeBC at very low

Figure 1. Monthly average values equivalent black carbon for (a) non-cloudy periods and (b) cloudy periods. Panel (c) shows the scavenged fraction as a ratio of the monthly average. The arithmetic mean (solid line), median (dotted) and geometric mean (dashed-dotted) are shown.

concentrations of eBC (see Fig. 2d) also explains why F_{eBC} can sometimes be above unity (see Fig. 1c and panel a-c in Fig. 2), since extremely small and thus uncertain concentrations are used to calculate F_{eBC} .” Yes, exactly. Fig. 2d indicates that

Figure 2. Revised main Figure 1: **The seasonal cycle of equivalent black carbon (eBC) in ambient and cloudy air (November 2015 until November 2019).** (a) Box plot of eBC concentration for ambient air (periods with visibility above 5 km) of both MAAP instruments sampling on the whole-air inlet. (b) Box plot of eBC concentration for cloudy air (periods with visibility below 1 km and GCVI in operation). MAAP1 represents the eBC values for cloud residuals (corrected for the GCVI sampling efficiency and enrichment factor), while MAAP2 represents the eBC values for whole-air (both cloud residuals and interstitial air). Note the different y-axis scale in panel a and b. (c) Scavenged fraction of eBC as a box plot. The monthly mean and median values are shown as solid and dashed line, respectively. The centre line of the box plot represents the median, while the triangles show the arithmetic mean of the distribution. The edges of the boxes are the quartile range and the whisker the range of the data (defined as 1.5 times from the nearest quartile). The grey numbers give the number of hourly values contained in each box.

individual FeBC values both <0 and >1 are found even at the highest eBC concentration. This doesn't surprise me, given the low concentrations. And just because values <0 or >1 are found doesn't mean that the results are inherently problematic. But it does mean that the authors need to actually consider the uncertainties in some formal manner. Here, the precision, and the relative precision, should be much more important than the accuracy.

We agree and have revised the analysis. Based on the work of Asmi et al. (2021), we now use 1-hour mean averages (for both MAAPs) and have determined the amount of data which is below the limit of detection (2.2 and 4.2%, respectively, see below). In addition, we determined the statistical significance of our observations for the entire and monthly data sets, which showed that the monthly data sets (i.e. cloud residual and whole-air eBC concentrations) are statistically significantly different, except for May and October, which is reasonable since the scavenged fractions are around 1 (see Figure 1c). We added this information to the revised manuscript (see detailed reply below).

Fig. 1 is a key figure in the manuscript, underpinning the entire story. I therefore am surprised to see that the authors have, it seems, discarded the vast majority of the scavenged fraction results determined from the median values. This is because there are really only two months (March/April) during which the whole-air eBC $>$ cloud residual eBC. This contrasts with the mean values, for which the whole-air eBC is generally \geq the cloud residual eBC. This raises a key question: which is more appropriate to use? The mean or the median? There are reasons to think that either is appropriate, although for a multi-year dataset such as this I would typically consider median values more robust, as mean values can be very sensitive to extreme events. So, to my previous question, how do the results change if the geometric mean values are used instead? Also, are the conclusions robust to the exclusion of extreme values? Are the distributions normal? (Most likely not, given the substantial mean-median differences.) Given the behavior of the medians, I need to be convinced that the conclusions based on the mean values are robust. Or put differently, the authors need to relate what they observe in the mean to how they interpret things in the median throughout much of the manuscript (i.e., based on typical seasonal behavior, not extremes. As it is, the authors' are focusing on the behavior of extreme events out of the year rather than on the typical behavior. They note as such on Page 2 but still this is an important consideration in terms of what any of this means for Arctic climate. It also has implications for how the authors consider data in Fig. 2, where they seem to include all data points, including those that they just said are basically unreliable for the median. I find this overall a major weakness.

We agree that we were not consistent in using the median or arithmetic mean and that the mean is skewed due to extreme events (since the data is indeed not normally distributed). The way how Figure 1 was presented in the original manuscript version was not ideal (the mean was more emphasised than the median). We have revised our main Figure 1 in the following way: We now show the data as monthly box plots of the 1-hour (reprocessed) mean values. We still show the mean values (as small grey triangles) but make clear in the text that these are driven by larger pollution transport events. We added to the first paragraph of Sect. 2: *"Note that arithmetic mean values in Fig. 1a are generally above the monthly median values due to the contribution of sporadic long-range transport events of polluted air to the site."*

In the revised main Figure 1, we also show the range of data as whiskers and have added the amount of data behind each box as grey numbers into the figure. This is a much more transparent way of presenting the entire data set, while the original messages remain unchanged: i) good agreement between both MAAP-instrument showing the annual cycle of eBC with a

95 maximum in spring due to the Arctic haze (panel a), ii) clear difference between eBC inside cloud residuals and whole-air, with larger differences in the winter months compared to the summer and fall (panel b), iii) this is also reflected in the scavenged fraction F_{eBC} which also shows the same seasonal behaviour if the mean or median values of F_{eBC} within the median of the box-plot or the median calculated using the median monthly concentrations of eBC (panel c) are used.

The Wilcoxon rank-sum test has been used to compare the hourly averages of the whole-air and cloud residual observations. The test was performed on the difference between the eBC concentrations of cloud residuals and whole-air measurements. 100 The test confirms that for all but two months (i.e. May and October) the null hypothesis can be rejected and that there is a statistically significant (s.s.) (p-value < 0.05) difference between the two sets of data, such that eBC whole-air > eBC cloud residual (see Fig. 3 within this reply letter). This shows nicely that the whole-air eBC > cloud residual eBC is not only the case for two months of the year (March/April); we know that the two populations are significantly different for multiple months, such that $F_{eBC} < 1$. May and October show a non-statistically significant difference between the whole-air eBC and cloud 105 residual eBC concentrations. May and October correspond to the two F_{eBC} peaks. We added to the revised manuscript the following sentence (end of 2nd paragraph in Sect. 2): *“The two sets of measurements (cloud residual and whole-air) present distributions that are significantly different, for all months of the year except May and October; using the Wilcoxon rank-sum test it was shown that the eBC concentrations for the whole-air inlet measurements were significantly greater than the eBC concentrations for the cloud residuals (not shown). The two sets of eBC measurements, for both May and October are not 110 significantly different and correspond to the two peaks in F_{eBC} ”*

The abstract notes that “An understudied aspect is the role of BC in the formation of low-level clouds in the Arctic.” To nit-pick, this study does not address this question. It addresses whether BC is found within cloud droplet residuals when considered on a mass basis. This does not say anything about what role the BC, as opposed to other soluble components, play in controlling the formation of low-level Arctic clouds. A more accurate statement that would tie to what the current study 115 shows is “An understudied aspect is the extent to which BC is present in low-level clouds in the Arctic.” Similarly, the comes out in the first paragraph, where the authors mention “the ability of aged BC to serve as cloud condensation nuclei.” Perhaps this is semantics and just a pet peeve on my part, but the question is ultimately whether the BC “serves” as a CCN or whether it is just along for the ride with the other, more soluble material the contributes to the aging process. In other words, take away the BC and would you still have the CCN? Similarly, the met conditions ultimately govern the ability of any particle to act as 120 a CCN, not just BC (first sentence, second paragraph). In part, I’m getting at the point of whether it is important that BC serve as CCN from a perspective of the clouds, or if it is important more for the lifetime of the BC and its deposition onto snow and ice. The authors might consider making this distinction in a more nuanced manner.

We agree and have changed the respective sentence in the abstract as suggested by the reviewer. Concerning the role of BC to act as CCN, we have replaced the word “BC” with the more general term “aerosol” in the first two sentences of the second 125 paragraph since these criteria for cloud droplet activation, as the reviewer correctly points out, are true for all particles and not only for BC. We added a sentence concerning the last point addressed by the reviewer about the more important aspect of BC lifetime (and deposition) compared to BC serving as CCN. *“BC in the Arctic still contributes less in terms of particle number and thus to CCN than other natural primary or secondary aerosol sources (Adachi et al., 2022), however, the incorporation of*

Figure 3. Histograms of the eBC concentrations during cloudy periods for the cloud residuals (blue) and the whole-air inlet measurements (red). Hourly averages of eBC concentrations are binned using even intervals of 0.5 eBC (ngm^{-3}). The subplots are divided according to month (as labelled in the top left-hand corner) and not year. The difference between the two distributions is compared and the result of the Wilcoxon rank-sum test is displayed in the legend, along with the corresponding p-value.

130 *BC into cloud particles is an important aspect since it determines the atmospheric lifetime of BC and its deposition onto snow or ice (Browse et al., 2012)."* The remaining 3 sentences of the same paragraph then address the aspects of BC to act as CCN and remain unchanged.

The conclusions regarding the scavenged fraction in Fig. 1c rely entirely on the mean values of eBC. Yet, in Fig. 2 only median values of the scavenged fraction are shown. Why this disconnect? For consistency shouldn't one use mean values?

Also, I do not think it is fair to say that the “whiskers show the range of observed eBC” when it is evident that the scales are set to cut off the full range of values.

We agree and have revised both figures. We now show the entire range of data (via the whiskers) in both figures. In the revised figure 2, we now also use and show the 1-h mean values of eBC (before we used 10-min values) and added two grey dashed lines at 0 and 1 to show the ideal range of F_{eBC} and that the medians are mostly within that range.

The authors show there is a relationship between the scavenged fraction and eBC, with some hint that this is an inverse relationship. Is there a relationship between eBC and cloud water content?

Yes, good point. Indeed, the eBC concentration within the cloud residuals is increasing with increasing CWC, while this is not observed for the corresponding eBC of the whole-air (see Figure 4 below), which explains why F_{eBC} is showing the behaviour (as shown in Fig. 2a in the main manuscript). This is also driven by the seasonality of the CWC at Zeppelin Observatory (see panel c). Clouds in the winter usually have much lower CWC compared to summer and are more optically thin (see panel d). Thus the seasonality of F_{eBC} , beside the temperature, is also driven by the seasonality of the CWC. We added this information to the revised manuscript and Figure S6 to the revised SI.

“The relationship of F_{eBC} versus CWC (Fig. 2a) is also reflected just in terms of eBC concentration (see Fig. S6a for all temperatures) meaning that more eBC is incorporated into the cloud particles with increasing CWC. However, this relationship is not seen for the whole-air eBC concentration versus CWC (see Fig. S6b), with even slightly higher eBC concentrations at low CWC. This again indicates that more eBC stays in the interstitial phase especially in the winter, when CWC is generally lower compared to the summer (see Fig. S6c and Fig. 1b and c).”

See also comment below on the interlinked relationships of F_{eBC} , CWC and the seasonal cycle of them.

The authors note that there are 2,207 hours of in-cloud data with visibility <1 km collected. In Fig. 2 it is indicated that there are $N = 709$ number of data points for scavenging fraction data. What is the relationship, then, between these 2,207 and 709 values? What constitutes a “data point?” And how does this relate to the statement in the methods that “The scavenged fraction of eBC, F_{eBC} , was then calculated using the monthly mean eBC values of the cloud residuals divided by the corresponding values of the whole-air.” There are only four years of data considered, so this would presumably mean a maximum of 48 “data points” where a data point is taken as a monthly mean.

In the original Figure 2, we used 10-min average for the binned data. In this case, e.g., 709 datapoints x 15 bin x 10 min = 1773 hours of data. This is less than 2207 hours of data because in this particular case the data was binned by temperature, which was not fully available for the entire GCVI data time series. The same issue of different temporal overlap holds when binning the data by updraft or CWC. The individual time series have slightly different overlaps. In the revised version with the reprocessed data we now use 1-hour averages throughout the work and mention the number of points contained in each box above the panel or as grey numbers (if not equally sized bins were used).

The wording “...using the monthly mean eBC values...” was misleading (it was just true for Figure 1c in the original manuscript), thanks for spotting this. We have replaced “monthly” with “hourly” in the revised manuscript.

The fewer number of “Data points” for the CWC and updraft periods in Fig. 2 versus the ambient temperature and eBC data points indicates that there was incomplete sampling. This is fine, and not surprising for a multi-year record. However, the

Figure 4. New Figure S10: **eBC concentrations binned by cloud water content (CWC) and annual cycle of CWC and visibility.** (a) eBC within cloud residuals vs. CWC. (b) eBC of whole-air (during cloudy periods) vs. CWC. (c) Monthly values of CWC. (d) Corresponding monthly values of visibility. The centre line of the boxes represents the median, while the extent of the boxes show the interquartile range. The whisker show the range of data (defined as 1.5 times the interquartile range from the nearest quartile). The number of 1-h mean values in each box is given above the panel (a-b) or as grey number within the panel (c-d).

170 authors might show as a supplemental the actual instrumental coverage to illustrate which of the 709 data points for Fig. 2b and Fig. 2c include the 590 CWC data points, and where there is not overlap. What sort of biases might this impart?

As mentioned above, we used 10-min values in the original Figure 2. If we use fully collocated data (CWC, eBC, temperature and updraft all have to be measured at the same time), we get, as expected, less data points per bin (see Figures 6 and 8 below), however, the overall trends and observations stay almost unchanged. Thus there is no substantial bias by using the most data as possible when binning with respect to CWC, temperature, eBC concentration or updraft.

175 As suggested, we have also updated Figure S11 in the SI (see Figure 7 below), now showing the actual data coverage in terms of hourly values (with the new reprocessed data) of eBC during cloud-free and cloudy periods. In this figure, we now

Figure 5. Revised Figure 2: **The scavenged fraction of eBC binned by cloud water content, ambient temperature, updraft and (whole-air) eBC concentration (a-d).** The exponential fits in (a) are shown for the 1-h mean values (orange dashed curve) and for the binned median values (red curve), respectively together with their corresponding 95% confidence intervals (shaded area). The corresponding fit coefficients are given in the legend together with their 95% confidence intervals. The number of approximate data points (1-h values) in each box are shown above each panel. The center line of the boxes represents the median, while the extend of the boxes shows the upper and lower quartile values. The whiskers indicate the range of observed F_{eBC} (defined as 1.5 times the interquartile range from the nearest quartile). The shading of the color of the box plots denotes the values of the x-axis. The grey dots show the underlying 1-hour mean values (panel a-c are limited to $F_{eBC} = \pm 4$ and panel d to ± 10 , respectively). The grey dashed horizontal lines are to guide the eye as the ideal range of F_{eBC} is between 0 and 1.

also show the reduced number of observations when collocating with other parameters such as cloud water content, ambient temperature and updraft.

The authors state on Page 1 that “The removal of BC from the atmosphere can occur through dry or wet deposition. Wet deposition describes the process of aerosol activation and subsequent removal by precipitation, also termed nucleation scavenging.” Here, I emphasize the phrase “subsequent removal by precipitation.” This work does not consider this key step. It only considers the fraction of BC that is detected in cloud residuals. Given the authors’ own definition of scavenging, I strongly

Figure 6. Same as Figure 2 in the revised manuscript but using **fully collocated** data (number of points per bin/box is given as n_{box} in the panel above).

encourage a revision to the title, abstract, terminology of FeBC (as the scavenging fraction), and really the language throughout to make clearer what they have actually characterized, which is the fraction of BC that is present in low-level clouds. This is not
 185 the same as the scavenging fraction. Should they want to refer to this as the “potential scavenging fraction” or some other such terminology I would be okay with this. But what they do not characterize is the scavenging. Scavenging must also consider the actual removal from the atmosphere term.

Here we have to kindly disagree with the reviewer. The term scavenging is often used to just describe the incorporation
 of particles into cloud droplets without including the actual removal process (most of the literature that uses CVI techniques
 190 (e.g., Schwarzenboeck et al., 2000; Sellegri et al., 2003; Glantz et al., 2003; Cozic et al., 2007; Mertes et al., 2007), as used here, but also e.g. in terms of cloud water measurements (e.g., Hitzenberger et al., 2003)). In most of the literature, the term does not include the actual removal from the atmosphere via precipitation. For example, Ohata et al. (2016) defines it as
“Incorporation of an aerosol particle into a water droplet (scavenging) in a cloud-and-precipitation system can occur via two distinct physical mechanisms: nucleation scavenging and impaction scavenging³. The former mechanism refers to an

Figure 7. Updated Figure S2: **Available observations.** Hours of observations during non-cloud periods (visibility > 5 km, red dashed curve) and during cloudy periods when the GCVI was in operation (visibility < 1 km). Shown are periods when both MAAP instruments measured eBC (blue solid line) and when corresponding auxiliary data was available for ambient temperature (blue dashed line), cloud water content (blue dashed dotted line) and updraft (blue dotted line).

195 *aerosol (cloud condensation nuclei: CCN) inducing formation of a cloud droplet in supersaturated water vapour and the latter mechanism refers to collision-coalescence of an aerosol and a water droplet. Scavenged aerosols are irreversibly removed from the atmosphere if the water droplets gravitationally fall and reach the ground.* and cites the standard text book by Pruppacher and Klett (“Microphysics of Clouds and Precipitation”). In chapter 17.4 of Pruppacher and Klett (1997 edition), it states: “As noted earlier, aerosol particles become incorporated into cloud drops by the mechanisms of nucleation and impact ion scavenging.” So here, just the pure incorporation of particles into cloud droplets is termed scavenging. However, we agree that it is not always used in a consistent manner, the activation into droplets and subsequent removal are sometimes not really separated or it is just assumed that the activated droplet will also precipitate. Even in Pruppacher and Klett (1997), in chapter 17.4.1 later on, the word “removal” is used again together with the term nucleation scavenging (“...this mode of aerosol removal is therefore termed nucleation scavenging.”).

205 However, we have revised the 3rd paragraph of the introduction and made clear that the term scavenging does not necessarily include the actual removal. The paragraph now reads: “BC can be incorporated into cloud droplets via scavenging, of which there are two different mechanisms: nucleation and impaction scavenging (Ohata et al., 2016; Pruppacher and Klett, 1997). Within this work, we will use the general term BC scavenging, since it is not possible to differentiate these two processes with the experimental techniques used here. However, it should be kept in mind that BC scavenging is dominated by nucleation scavenging (Ohata et al., 2016). Once the BC is scavenged, it can then be removed completely from the atmosphere as a result of precipitation. It should be made clear that the following text refers to the incorporation of BC into hydrometeors, and does not concern itself with the removal of scavenged aerosol.”

Page 1: How do “large uncertainties” translate to an estimate “missing?” To me this does not logically follow. The large uncertainties mean that estimates of the impact are uncertain. Not that such estimates are “missing.”

215 We agree and have replaced the last word of this sentence with “still uncertain”, it now reads: *“The large uncertainties surrounding the impacts of the indirect effects of aerosols on clouds mean that an estimate of the net impact of BC in the Arctic is still uncertain.”*

Page 2: The authors note that “However, the seasonal cycle and magnitude of eBC as shown in Fig. 1a agrees very well (also with respect to magnitude) with the observed seasonal cycle using more recent and advanced observations by Sinha et al.” I do not follow this, in particular the “more recent” statement. First, the Sinha paper is from 2017 yet the work here encompasses measurements from 2015-2019. So how is Sinha et al. “more recent?” Also, how are the Sinha et al. measurements “more advanced?” They too use filter-based absorption measurements. A MAAP, as used here, is arguably more “advanced” than a PSAP or COSMOS considered by Sinha et al.

We agree that we were not very clear with our wording here. The “more recent” was in comparison to Eleftheriadis et al. (2009), who reported observations from the years 1998-2007, while Sinha et al. (2017) reported results from 2006-2015 for the same site. The “more advanced” related to the fact that Sinha et al. (2017) performed more detailed analysis with respect to the site-specific MAC value.

We changed the sentence which now reads: *“However, the seasonal cycle and magnitude of eBC as shown in Fig. 1a agrees very well (also with respect to magnitude) with the observed seasonal cycle reported by Sinha et al. (2017) for the years 2006 to 2015.”*

Page 2: The authors note that “During cloudy periods (Fig. 1b), the whole-air eBC, consisting of activated and interstitial (non-activated) particles, follows the same typical seasonality...” and then report mean values. But in the median the values seem to be basically, or very near zero for all cloudy periods with the exception of two months. Why focus on the mean and not the median here? As to a previous comment, wouldn't the means emphasize extreme events?

As mentioned above, we have reprocessed the data and now show a box plot for our main Figure 1. We agree that the mean is too much influenced by extreme events and have replaced these values by the corresponding median and the upper and lower quartile range. The sentence now reads: *“...follows the same typical seasonality, with a clearly lower overall concentration than during non-cloudy periods (median and interquartile range, IQR, for the entire four years: 6.17 (IQR: 2.71 - 13.95) ngm⁻³ for non-cloudy periods and 2.07 (IQR: 0.73 - 4.66) ngm⁻³ for cloudy periods, respectively).”* We also replaced the mean values in the last sentence with the corresponding median values and IQR. The sentence now reads; *“The concurrently measured eBC concentration of the cloud residuals shows lower values in winter (e.g. in January with whole-air eBC concentration of 2.8 (1.3 - 6.7) ngm⁻³ (median and IQR), compared to cloud-residual eBC concentration of 0.1 (-0.1 - 0.6) ngm⁻³), indicating that not all eBC have been activated or taken up by cloud particles. Later in spring and summer, the eBC concentrations of the cloud residuals are on average similar to the whole-air measurements (median between 0.6 to 8.0 ngm⁻³ between April and June), revealing that most of the eBC was activated into cloud droplets.”*

The comparison to Heintzenberg and Leck (1994) would be strengthened if the authors here used their higher time-resolution measurements to calculate the equivalent in-cloud/out-of-cloud fraction as determined by H&L. Are different results obtained? This should be as straightforward as calculating the ratio between the whole-air values from MAAP 2 in Fig. 1b and Fig. 1a. Is there a seasonal variation that can explain the H&L results?

250 A comparison with H&L, using our higher time-resolution measurements, can be easily done. H&L take the average of
INT/OOC ¹ for two periods 16th May - 15th October (winter) and 16th October to 15th May. If we were to calculate the
averages (median) for these two periods (but for the collocated data), we receive a scavenged fraction of 0.42 for winter and
0.53 for their summer period, respectively. We added this to the revised manuscript with the following: “*If we average for*
the same monthly periods as Heintzenberg and Leck (1994) (using the median), we receive scavenged fractions of 0.42 for
255 *winter and 0.53 for summer period, respectively. This is substantially lower and probably can be explained that the fractions*
determined by Heintzenberg and Leck (1994) ...”. If we calculate the ratio using non-collocated data (so dividing cloudy
events by non-cloudy events, as Heintzenberg and Leck did), we receive FeBC values of 0.10 for summer and 0.13 for winter,
respectively. This is substantially lower but not surprising since we have shown before that the out-of-cloud eBC values are
generally higher than in-cloud eBC values. In addition (and due to the comment from reviewer 2), we added the fact that
260 Heintzenberg and Leck did not use a cloud sensor to detect the presence of clouds but indirectly inferred it from their aerosol
data, which could be another source of uncertainty. In addition, we clarified that the study determined the ratio between
interstitial and out-of-cloud eBC concentrations. We added: “... *using an optical filter-based light absorption technique. As*
such, the study determined the ratio between interstitial (during cloudy periods) and PM₁ during out-of-cloud periods, which
can be converted to F_{eBC} (similar as done by Cozic et al. (2007)) by assuming that the residual eBC concentration equals the
265 *eBC concentration of PM₁ minus the interstitial value. They did not use a cloud sensor to determine the in-cloud periods but*
used a data reduction scheme to infer the presence of cloud at the station.”

Besides the study by Heintzenberg and Leck (1994), the work by Adachi et al. (2022) is also supporting our findings with
respect to the observed seasonality. In their study, single-particle analysis together with an elemental composition analysis of
whole-air aerosols and cloud residuals were performed at the same site.

270 We added to the revised manuscript to the same section (end of paragraph 3 of Sect. 2): “*Adachi et al. (2022) observed, at the*
same site, the smallest ratio of residual relative to ambient carbonaceous particles during the winter season, thus observing
similar seasonality.”

How would the authors define a “clear dependence” (or lack thereof) as they use it with reference to the updraft velocity-
scavenged fraction relationship (Page 4) or the FeBC-CWC relationship, or any other relationship? Is there some statistical
275 metric?

This is a very good point and we have tested various approaches. The FeBC-CWC relationship can e.g. be shown by plotting
the logarithm of both values (which then shows a linear relationship). However, we have decided to use the same approach
as Cozic et al. (2007) (to allow a better comparison) and added an exponential fit through (i) the 1-h mean values and (ii)
through the median of the binned data. Both show a clear increase of FeBC with increasing CWC with reasonable confidence
280 intervals (CI). We added the fits as well as the underlying 1-h data to the revised Figure 2. We also added the exponential fit
to the FeBC-CWC relationship for clouds above -5°C (as done by Cozic et al, see revised SI Fig. S5a). The fit and their CI's
show a reasonable functional relationship. We also tested various fits for the FeBC-T relationship (e.g. sigmoidal functions),

¹Here, we use the whole-air values for their OOC (out-of-cloud) and calculate INT (interstitial) by subtracting the whole-air-values by the concentration
in the residuals.

however, the CI's of the fit coefficients were unreasonably large. However, by just comparing the 1-h and the binned FeBC data for temperatures below and above -5°C , it is clear that two distinct populations are present (i.e. FeBC at lower temperatures are clearly lower compared to higher temperatures, which show much more variability and are on average higher). The FeBC-updraft and FeBC-eBC data clearly show no relationships. We should maybe mention that we tested simple linear relationships: while FeBC-CWC and FeBC-T are positively correlated (Pearson correlation coefficient (r) = 0.62 and 0.72, respectively), the FeBC-updraft and FeBC-eBC showed no significant correlation (r =-0.56 and 0.17, respectively). We updated Figure 2 (in the main text) and S5 (in the supplement, showing the FeBC-CWC for temperatures above and below -5°C and added the following text to the revised manuscript (2nd paragraph ind Sect.3): *“Two fits for both 1-h values and the binned median values of F_{eBC} , respectively, are added to Fig. 2a, showing the exponential increase of F_{eBC} with increasing CWC (see legend for fit coefficients and confidence intervals).”* We also added to the following sentence to the comparison with the findings by Cozic et al.: *“..., while F_{eBC} -values of around 1 (median) at Zeppelin Observatory are already reached at around 0.15 gm^{-3} CWC.”* We also modified the 1st sentence of the 3rd paragraph and instead of dependency talk about two different populations or regimes: *“The observed F_{eBC} -values also show two distinct regimes or populations with respect to the ambient temperature, with clearly lower scavenged fractions below around -5°C .”*

The authors compare their results to those from Jungfraujoch, another remote site. However, the mean eBC concentrations in Jungfraujoch are about an order of magnitude higher than here. So, while such measurements are still challenging, the importance of a careful uncertainty assessment is different. The differences in the FeBC values between the studies could be, ultimately, negligible once uncertainties are accounted for. I am skeptical that any of the relationships shown in the current study are “real” and not simply a function of the parameter of interest (CWC or air temperature) (anti)correlating with eBC. I need to be convinced that the relationships observed are real.

With our revised Figure 2, it should be clear that FeBC and CWC show a relationship (which is illustrated with a respective fit, see comments above). However, we agree with the reviewer that these relationships (FeBC vs CWC and FeBC vs temperature) are interlinked and also linked to the seasonal cycle. We have further looked into this by plotting the seasonal cycle of CWC (and visibility) as shown in the new Fig. S6. Clouds show indeed a much lower CWC in the colder months (November until February, see Fig. S6c) when FeBC is also lower. The sampled clouds were also optically thinner (see higher visibility values in Fig. S6d).

We added a new paragraph covering this aspect to the last part of Sect.3: *“The relationships of F_{eBC} vs. CWC and F_{eBC} vs. temperature are interlinked with the seasonality of clouds observed at Zeppelin Observatory. In the winter, the observed clouds were in general thinner with on average lower CWC (or higher visibility) compared to the summer months (see Fig. S6 c and d). Therefore, the seasonality of lower F_{eBC} in the winter months (see Fig. 1c) can be explained by the fact that CWC is generally lower in the winter (Fig. 2a and S6c) and in parts due to mixed-phase cloud processes (Fig. 2b). The separation between both effects is only possible with collocated detailed cloud microphysical measurements which includes the separation between droplets and ice crystals.”*

On page 5 the authors state at the end of a sentence that “for cloudy periods, eBC is removed regardless of the surface type.” It is not clear how this follows from the previous part of the sentence. Where does the “removal” aspect come from?

We agree with the comment here, it is not clearly explained what we mean, and has been re-written.
From Figure S2, we observe differences in the cloudy and non-cloudy data sets. For observations made during non-cloudy
320 periods, high eBC concentrations correspond to air masses which had spent more time over continental surface types, whilst
low eBC concentrations coincide with air masses of a more marine influence. However, when observations are made in cloudy
conditions (i.e. in-cloud), there is no such dependency on surface type; air masses influenced more by marine surface types
have a similar eBC concentration as air masses influenced more by the continent. An explanation for this could be that the acti-
vated/scavenged fraction is much greater for air masses more influenced by marine surface types, due to the higher proportion
325 of clouds present (see Fig. S3).

We altered the text, such that it reads as follows: “... *Figure S2 (in the SI) shows that for non-cloudy eBC, measured via the
whole-air inlet, there is a clear dependence on the time that coinciding air masses spend over continental source regions,
whereas for cloudy periods eBC concentrations of cloud residuals are much lower and there is no relation to the time air
masses spend over continental regions; observations of cloud residuals coinciding with air masses influenced more by marine
330 surface types have a similar eBC concentration as air masses influenced more by the continent....*”

2 Comments and reply to Reviewer #2

Review of “Black carbon scavenging by low-level Arctic clouds” by Ziegler et al. (read abstract and introduction one more
time)

Below I address the questions raised by Nature Communications about the submitted manuscript. “Q” represents the ques-
335 tion, and “A” represents the answer.

Q1: What are the noteworthy results?

A1: The authors present the first direct long-term observational study of the scavenging of black carbon (BC) by clouds.
The authors collected an impressive four years of data allowing for seasonal cycles and dependence on several factors to be
elucidated. From these studies, the authors conclude that the scavenging efficiency of BC by clouds was positively correlated
340 with the cloud water content and depended on the ambient temperature.

Q2: Will the work be significant to the field and related fields?

A2: Yes. This unique dataset will be useful for model improvements or model validation exercises. In other words, these
measurements should lead to better models and predictions of the effect of BC on the Arctic climate.

Q3: How does it compare to the established literature? If the work is not original, please provide relevant references.

345 A3: As indicated by the authors, a previous study by Cozic et al. 2007, ACP, showed that the scavenging efficiency of BC was
positively correlated with the cloud water content and depended on ambient temperature. The previous study by Cozic et al.
was carried out at a European mountain site. The current study shows a similar behavior but for the Arctic region. The authors
stated in the Conclusion that “the scavenging fraction of BC increased as expected with the increasing cloud water content.” In
other words, the observations are expected but critical to confirm for the Arctic, which is experiencing rapid climate change.

350 Q4: Does the work support the conclusions and claims, or is additional evidence needed? - Are there any flaws in the data analysis, interpretation, and conclusions? - Do these prohibit publication or require revision? - Is the methodology sound? Does the work meet the expected standards in your field?

A4: The methodology is sound, and the work is of very high standards. The manuscript is also very well written, and the work supports the conclusions and claims.

355 Below are additional comments for the authors to consider.

1) Page 2, last paragraph. “The fractions determined by Heintzenberg and Leck did not use temporarily collocated measurements and therefore did not consider differences in air origin during cloudy periods compared to non-cloud period, where the overall concentrations of eBC are different.” I had to go and look up the paper by Heintzenberg and Leck to understand the difference between these measurements and the new results by Zieger et al. The authors may want to rewrite this paragraph to
360 clarify it for the non-expert.

Heintzenberg and Leck did not sample cloud droplets and ambient aerosol simultaneously as part of the same set-up, instead they sampled interstitially i.e. aerosol inside boundary layer clouds (INT) and out-of cloud (OOC) groups over a longer time and averaged the results of each. They also used a different cloud-detection scheme. With the comments from reviewer 1 in mind, we added this information and modified the entire paragraph to:

365 *“Heintzenberg and Leck (1994) determined eBC fractions in the early 1990s at the same site, comparing eBC concentrations within and outside clouds (using a PM₁ inlet) using an optical filter-based light absorption technique. As such, the study determined the ratio between interstitial (during cloudy periods) and PM₁ during out-of-cloud periods, which can be converted to F_{eBC} (similar as done by Cozic et al. (2007)) by assuming that the residual eBC concentration equals the eBC concentration of PM₁ minus the interstitial value. They did not use a cloud sensor to determine the in-cloud periods but used a data reduction
370 scheme to infer the presence of cloud at the station. Heintzenberg and Leck (1994) found for summer (mid-May to mid-October 1990-1992) and winter (mid-October to mid-May 1990-1992) average F_{eBC}-values of 0.81 and 0.77, respectively. If we average for the same monthly periods as Heintzenberg and Leck (1994), we receive scavenged fractions of 0.42 for winter and 0.53 for summer period, respectively. This is substantially different and probably can be explained by the different set-up, the difference cloud-detection scheme and that the fractions determined by Heintzenberg and Leck (1994) did not use
375 temporarily collocated measurements and therefore did not consider differences in air origin during cloudy periods compared to non-cloudy periods, where the overall concentrations of eBC are different (cf. Fig. 1).”*

2) Beginning of Section 4. “Therefore, the source region of eBC in and outside cloud periods is expected to be different. This is further demonstrated in Fig. 3 showing that cloudy periods are characterized by very low eBC concentrations of mainly marine origin”. What features in Figure 3 lead to this conclusion? From Fig. 3a, the median eBC concentration at trajectory
380 endpoints is highest over land. I agree there is more red color over land in 3b vs. 3a, but land vs. marine in 3a is not clear to me.

We agree with the comments made by the referee. The previous statement in the manuscript is too strong a statement; we have removed the statement that “cloud periods are characterized by ... eBC concentrations of mainly marine origin.” However, it still remains the case that the source regions for eBC presented in Fig. 3 are different. We can refer to Fig. S2 to

385 help interpret these results. In general, Fig. 3a is composed of more data points, and as such the reach of the back trajectories is more expansive. For Fig. 3a, the region which contributes the most eBC (i.e. central Eurasia) is not present. Instead, the average grid cell is more homogenised (i.e. eBC concentrations are more similar), with the exception of a few marine and terrestrial “hotspots” e.g. the Norwegian coastline.

Figure 8. New Figure S11 showing the effect on the seasonal cycle on the scavenged fraction when excluding high updraft values (replacing the dashed and dashed-dotted lines from Figure 1c in the original submitted manuscript): **The annual cycle of the scavenged fraction of eBC.** Scavenged fraction of eBC as a box plot for all data, and for vertical wind values below 1 ms^{-1} and 0.5 ms^{-1} , respectively. Shown are monthly mean and median values as solid and dashed line, respectively. The centre line of the boxes represents the median, while the extent of the boxes show the interquartile range. The whisker show the range of data (defined as 1.5 times the interquartile range from the nearest quartile). All box plots contain hourly mean values. The numbers in blue, light blue and grey give the number of hourly value contained in each month for each data class (see legend).

3 Further changes

390

- During the revision of Figure 1 in the main manuscript (now showing monthly box plots, see comment(s) above), we decided to show the effect of removing high updraft values in a separate figure (in the SI, see Fig. S4 and Fig. 8 within this reply letter) to avoid a too messy main figure. We adapted the corresponding text (end of 3rd paragraph in Sect. 2): “Figure S4 (in the supplementary information) shows that higher values of F_{eBC} are not significantly driven by increased updraft values, as removing F_{eBC} values with high updraft values (above 1 ms^{-1} and 0.5 ms^{-1} , respectively) keeps the monthly distribution of F_{eBC} values almost unchanged (see also Fig. 2c in next section).”

395

- We made minor changes to language after a final editorial read. We also added information on the definition of the whiskers and boxes of the boxplots to the respective captions.
- We reordered the figures in the SI so that the order follows the mentioning within the main text.

400 References

- Adachi, K., Tobo, Y., Koike, M., Freitas, G., Zieger, P., and Krejci, R.: Composition and mixing state of Arctic aerosol and cloud residual particles from long-term single-particle observations at Zeppelin Observatory, Svalbard, *Atmos. Chem. Phys.*, 22, 14 421–14 439, <https://doi.org/10.5194/acp-22-14421-2022>, <https://acp.copernicus.org/articles/22/14421/2022/>, 2022.
- Asmi, E., Backman, J., Servomaa, H., Virkkula, A., Gini, M. I., Eleftheriadis, K., Müller, T., Ohata, S., Kondo, Y., and Hyvärinen, A.: Absorption instruments inter-comparison campaign at the Arctic Pallas station, *Atmos. Meas. Tech.*, 14, 5397–5413, <https://doi.org/10.5194/amt-14-5397-2021>, <https://amt.copernicus.org/articles/14/5397/2021/>, 2021.
- Browse, J., Carslaw, K. S., Arnold, S. R., Pringle, K., and Boucher, O.: The scavenging processes controlling the seasonal cycle in Arctic sulphate and black carbon aerosol, *Atmos. Chem. Phys.*, 12, 6775–6798, <https://doi.org/10.5194/acp-12-6775-2012>, <https://acp.copernicus.org/articles/12/6775/2012/>, 2012.
- 410 Cozic, J., Verheggen, B., Mertes, S., Connolly, P., Bower, K., Petzold, A., Baltensperger, U., and Weingartner, E.: Scavenging of black carbon in mixed phase clouds at the high alpine site Jungfraujoch, *Atmos. Chem. Phys.*, 7, 1797–1807, <https://doi.org/10.5194/acp-7-1797-2007>, <https://acp.copernicus.org/articles/7/1797/2007/>, 2007.
- Eleftheriadis, K., Vratolis, S., and Nyeki, S.: Aerosol black carbon in the European Arctic: Measurements at Zeppelin station, Ny-Ålesund, Svalbard from 1998–2007, *Geophys. Res. Lett.*, 36, L02 809, <https://doi.org/10.1029/2008GL035741>, 2009.
- 415 Glantz, P., Noone, K. J., and Osborne, S. R.: Scavenging efficiencies of aerosol particles in marine stratocumulus and cumulus clouds, *Quarterly Journal of the Royal Meteorological Society: A journal of the atmospheric sciences, applied meteorology and physical oceanography*, 129, 1329–1350, 2003.
- Heintzenberg, J. and Leck, C.: Seasonal variation of the atmospheric aerosol near the top of the marine boundary layer over Spitsbergen related to the Arctic sulphur cycle, *Tellus B*, 46, 52–67, 1994.
- 420 Hitztenberger, R., Giebl, H., Petzold, A., Gysel, M., Nyeki, S., Weingartner, E., Baltensperger, U., and Wilson, C. W.: Properties of jet engine combustion particles during the PartEmiss experiment. Hygroscopic growth at supersaturated conditions, *Geophys. Res. Lett.*, 30, 1779, <https://doi.org/10.1029/2003GL017294>, 2003.
- Mertes, S., Verheggen, B., Walter, S., Connolly, P., Ebert, M., Schneider, J., Bower, K., Cozic, J., Weinbruch, S., Baltensperger, U., et al.: Counterflow virtual impactor based collection of small ice particles in mixed-phase clouds for the physico-chemical characterization of tropospheric ice nuclei: Sampler description and first case study, *Aerosol Sci. Technol.*, 41, 848–864, 2007.
- 425 Ohata, S., Moteki, N., Mori, T., Koike, M., and Kondo, Y.: A key process controlling the wet removal of aerosols: new observational evidence, *Scientific reports*, 6, 1–9, 2016.
- Petzold, A. and Schönlinner, M.: Multi-angle absorption photometry—a new method for the measurement of aerosol light absorption and atmospheric black carbon, *J. Aerosol Sci.*, 35, 421–441, 2004.
- 430 Pruppacher, H. R. and Klett, J. D.: *Microphysics of clouds and precipitation*, Springer, 1997.
- Schwarzenboeck, A., Heintzenberg, J., and Mertes, S.: Incorporation of aerosol particles between 25 and 850 nm into cloud elements: measurements with a new complementary sampling system, *Atmospheric research*, 52, 241–260, 2000.
- Sellegri, K., Laj, P., Dupuy, R., Legrand, M., Preunkert, S., and Putaud, J.-P.: Size-dependent scavenging efficiencies of multicomponent atmospheric aerosols in clouds, *Journal of Geophysical Research: Atmospheres*, 108, 2003.
- 435 Sinha, P., Kondo, Y., Koike, M., Ogren, J., Jefferson, A., Barrett, T., Sheesley, R., Ohata, S., Moteki, N., Coe, H., et al.: Evaluation of ground-based black carbon measurements by filter-based photometers at two Arctic sites, *J. Geophys. Res.*, 122, 3544–3572, 2017.

Reviewer #1 (Remarks to the Author):

I appreciate the authors thoughtful responses and can recommend this work for publication.